# DepNeCTI: Dependency-based Nested Compound Type Identification for Sanskrit

**Jivnesh Sandhan**[†*], **Yaswanth Narsupalli**[‡*], **Sreevatsa Muppirala**[‡],
**Sriram Krishnan**[§], **Pavankumar Satuluri**[‖], **Amba Kulkarni**[§] and **Pawan Goyal**[‡]

[†]UC Berkeley, [‡]IIT Kharagpur, [§]University of Hyderabad and [‖]IIT Roorkee

jivneshsandhan@gmail.com, yasshu.yaswanth@gmail.com, pawang@cse.iitkgp.ac.in

## Abstract

Multi-component compounding is a prevalent phenomenon in Sanskrit, and understanding the implicit structure of a compound's components is crucial for deciphering its meaning. Earlier approaches in Sanskrit have focused on binary compounds and neglected the multi-component compound setting. This work introduces the novel task of nested compound type identification (NeCTI), which aims to identify nested spans of a multi-component compound and decode the implicit semantic relations between them. To the best of our knowledge, this is the first attempt in the field of lexical semantics to propose this task.

We present 2 newly annotated datasets including an out-of-domain dataset for this task. We also benchmark these datasets by exploring the efficacy of the standard problem formulations such as nested named entity recognition, constituency parsing and seq2seq, etc. We present a novel framework named DepNeCTI: **Dep**endency-based **Ne**sted **C**ompound **T**ype **I**dentifier that surpasses the performance of the best baseline with an average absolute improvement of 13.1 points F1-score in terms of Labeled Span Score (LSS) and a 5-fold enhancement in inference efficiency. In line with the previous findings in the binary Sanskrit compound identification task, context provides benefits for the NeCTI task. The codebase and datasets are publicly available at: https://github.com/yaswanth-iitkgp/DepNeCTI

## 1 Introduction

A compound is defined as a group of entities functioning as a single meaningful entity. The process of identifying the implied semantic relationship between the components of a compound in Sanskrit is known as Sanskrit Compound Type Identification (SaCTI) (Sandhan et al., 2022a) or Noun Compound Interpretation (NCI) (Ponkiya et al., 2021,

2020). Within the literature, the NCI problem has been approached in two ways, namely, classification (Dima and Hinrichs, 2015; Fares et al., 2018; Ponkiya et al., 2021) and paraphrasing (Lapata and Keller, 2004; Ponkiya et al., 2018, 2020).

In Sanskrit literature, particularly in poetry, the use of multi-component compounds is ubiquitous (Kumar, 2012). According to the Digital Corpus of Sanskrit, more than 41% of compounds contain three or more components (Krishna et al., 2016). However, earlier approaches focus solely on binary compounds and fail to address the complexities inherent in multi-component compounds adequately. Thus, we propose a new task, the Nested Compound Type Identification (NeCTI) Task, which focuses on identifying nested spans within a multi-component compound and interpreting their implicit semantic relationships. Figure 1 illustrates an example of the NeCTI task, highlighting nested spans and their associated semantic relations using distinct colors.

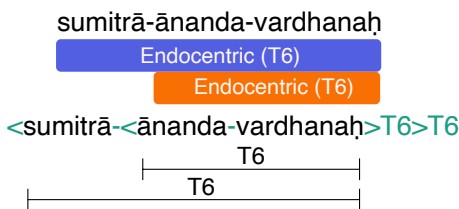

Figure 1: Illustration of the NeCTI task for the multi-component compound *sumitrā-ānanda-vardhanāḥ* (Translation: Sumitrā-delight-enhancer), highlighting nested spans and their associated semantic relations using distinct colors.

The NeCTI task presents multiple challenges: (1) The number of potential solutions for a multi-component compound grows exponentially as the number of components increases. (2) It often relies on contextual or world knowledge about the entities involved (Krishna et al., 2016). Even if a multi-component compound shares the same components

---
[*] denotes the first two authors contributed equally.

and final form, the implicit relationship between the spans can only be deciphered with the aid of available contextual information (Kulkarni and Kumar, 2013; Krishna et al., 2016). For instance, as de-

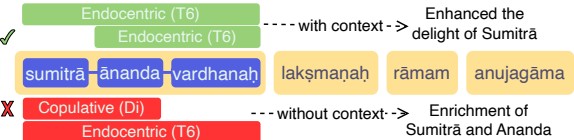

Figure 2: Illustration of the multi-component compound *sumitrā-ānanda-vardhanāḥ* (Translation: Sumitrā-delight-enhancer) with two valid parses depicted in green and red. The two parses correspond to two potential meanings. The green solution represents the correct interpretation within the provided context.

picted in Figure 2, the multi-component compound *sumitrā-ānanda-vardhanāḥ* (Translation: Sumitrā-delight-enhancer) can have two valid but distinct solutions, leading to different meanings. Resolving ambiguity to select the correct solution requires reliance on the provided context. Consequently, downstream Natural Language Processing (NLP) applications for Sanskrit, such as question answering (Terdalkar and Bhattacharya, 2019) and machine translation (Aralikatte et al., 2021), often exhibit sub-optimal performance when encountering compounds. Hence, the NeCTI task serves as a preliminary requirement for developing robust NLP technology for Sanskrit. Moreover, this dependency on contextual information eliminates the possibility of storing and conducting a lookup to identify the semantic types of nested spans.

Previous approaches (Kulkarni and Kumar, 2013; Krishna et al., 2016; Sandhan et al., 2019) addressing SaCTI have predominantly focused on binary compounds, neglecting the consideration of multi-component compounds. In multi-component compounds, the components exhibit semantic relationships akin to dependency relations, represented as directed labels within the dependency structure, which also facilitate the identification of the compound's headword through the labels directed towards it. Consequently, dependency formulation enables the simultaneous identification of both the structure or constituency span and the compound types. Thus, we propose a novel framework (§ 4) named DepNeCTI: Dependency-based Nested Compound type Identifier (§3). In summary, our contributions can be outlined as follows:

- We introduce a novel task called Nested Compound Type Identification (§ 2).

- We present 2 newly annotated datasets and provide benchmarking by exploring the efficacy of various standard formulations for NeCTI (§ 5).

- We propose a novel framework DepNeCTI: Dependency-based Nested Compound type Identifier (§ 4), which reports an average 13.1 points F1-score in terms of LSS absolute gain and 5-fold enhancement in inference efficiency (§ 6) over the best baseline.

- We publicly release the codebase of DepNeCTI and benchmarked baselines, along with newly annotated datasets for the NeCTI task.

## 2 Problem Formulation

The objective of the NeCTI task is to detect nested spans within a multi-component compound and decipher the implicit semantic relations among them. Our study focuses exclusively on this task and does not address the compound segmentation problem. It is assumed that the segmented components of the multi-component compound are already available. To obtain the segmentation of a compound, we rely on established resources such as the rule-based shallow parser (Goyal and Huet, 2016) or existing data-driven segmentation systems designed explicitly for Sanskrit (Hellwig and Nehrdich, 2018; Sandhan et al., 2022b).

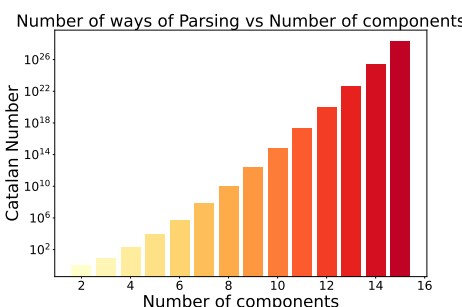

Figure 3: Illustration of a number of ways (as per Catalan number in log scale) in which a multi-component compound can be parsed. Our dataset contains compounds that have a maximum of 16 components.

**Complexity of NeCTI Task:** As the number of components in a multi-component compound increases, the number of possible parses grows exponentially. Our dataset encompasses multi-component compounds ranging from 2 to 16 components. Figure 3 visually depicts the exponential growth of possible parses with increasing component count. The parsing of a compound word with

$n + 1$ components can be likened to the problem of fully parenthesizing $n + 1$ factors in all possible ways (Kulkarni and Kumar, 2011). Thus, the total number of parse-options for a multi-component compound word with $n + 1$ components corresponds to the Catalan Number $C_n$, where $n \geq 0$ (Huet, 2009).

$$C_n = \frac{1}{n+1}\binom{2n}{n} = \frac{(2n)!}{(n+1)!n!}$$

where $C_n$ represents the $n^{th}$ Catalan number, $\binom{2n}{n}$ is the binomial coefficient, and ! denotes factorial. Finally, the compatibility rules derived from Pāṇinīan grammar (Paṇini, 500 BCE) and contextual information are needed to disambiguate multiple possibilities.

Formally, in a given sentence $X = \{x_1, x_2, ..., x_M\}$ with $M$ tokens, let $x_p$ ($1 \leq p \leq M$) denote an $N$-component compound. It is worth noting that $X$ may contain multiple instances of multi-component compounds. A valid solution corresponds to a full paranthesization of these $N$ components; let $P_N$ encompass all valid solutions for fully parenthesizing $N$ factors, satisfying the cardinality $|P_N| = C_{N-1}$, where $C_{N-1}$ represents the Catalan number. A valid solution for an $N$-component compound consists of $N-1$ nested spans. The NeCTI system produces an output represented as a list of $N-1$ tuples for $x_p$, given by $Y_p = \{[I_1^H, I_1^T, T_1], ..., [I_{N-1}^H, I_{N-1}^T, T_{N-1}]\}$, such that $Y_p \in P_N$. $I_i^H$ and $I_i^T$ denote the head and the tail indices, respectively, of the $i^{th}$ span, and $T_i$ corresponds to the label assigned to the respective span.

**How different is NeCTI compared to the Nested Named Entity Recognition (NNER) task?** NNER is a component of information extraction that aims to identify and classify nested named entities within unstructured text, considering their hierarchical structure. In contrast, the NeCTI task focuses on identifying nested spans within a multi-component compound and decoding their implicit semantic relations. These tasks have several key differences: (1) NeCTI operates at the intra-word level, whereas NNER operates at the inter-word level, considering entities across a phrase. (2) In NeCTI, a multi-component compound can have multiple possible parses, requiring disambiguation through contextual cues and incorporating insights from Pāṇinīan grammar to address incompatibil-

ities. Conversely, we could not find discussions related to these aspects in NNER literature. (3) NeCTI benefits from prior knowledge of the compound's location and segmented components. In contrast, the NNER task involves the additional challenge of identifying the location of entities within the text. Consequently, leveraging existing NNER frameworks for NeCTI is not straightforward due to their inability to provide explicit support for providing the location of compounds. Therefore, NeCTI presents unique characteristics and challenges that differentiate it from the NNER task, requiring specialized approaches tailored to its specific requirements.

**How different is NeCTI compared to the Multi-Word Expressions (MWE)?** MWEs encompass various categories such as idioms (e.g., *kick the bucket*), named entities (e.g., *World Health Organization*), and compounds (e.g., *telephone box*), etc. Sanskrit compounds exhibit similarities to multi-word expressions, particularly multi-word compounds (nominal, noun, and verb), with respect to characteristics like collocation of components based on semantic relations between them and strict preference for ordering of the components.

Multi-word compounds in various languages involve adjacent lexemes juxtaposed with potential semantic relations. In contrast, Sanskrit compounds are intuitively constructed based on the semantic compatibility of their components. Additionally, in Sanskrit, compounds always appear as a single word, requiring mandatory *sandhi* (euphonic transformations) between their components. Conversely, certain categories of MWEs (idioms, complex function words, verb-particles, and light-verbs) have relatively fixed structures, predominantly with components separated by spaces. Furthermore, nesting within multi-word expressions is considered syntactic overlaps, while the nested structures of Sanskrit compounds result from successive combinations of components based on semantic relations, thereby clearly distinguishing Sanskrit compounds from MWEs.

## 3 Why NeCTI as a Dependency Parsing Task?

The decision to formulate NeCTI as a dependency parsing task is driven by several considerations. Compounds with more than two components are typically formed through successive binary combinations, following specific semantic relations. This

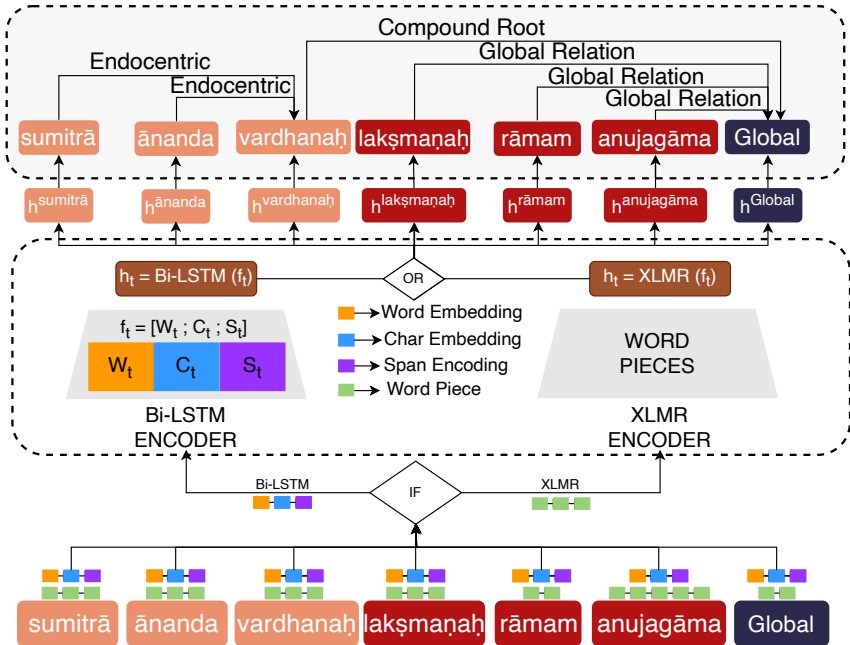

Figure 4: Illustration of DepNeCTI with an example "*sumitrā-ānanda-vardhanaḥ lakṣmaṇaḥ rāmam anujagāma*" (Translation: "Lakṣmaṇa, the one who enhanced the delight of Sumitrā, followed Rāma") where '*sumitrā-ānanda-vardhanaḥ*' is a multi-component compound word. We assume prior knowledge of compound segmentation and treat the individual components of multi-component compounds as separate words. That means the compound and its components are known apriori. However, the associations of the components, i.e. spans, are not known apriori. We propose two variants of DepNeCTI depending on the choice of encoder: DepNeCTI-LSTM and DepNeCTI-XLMR. To inform DepNeCTI-LSTM about compound (highlighted with ▆ color) and non-compound (highlighted with ▆ color) tokens, we employ span encoding. The span encoding uses two randomly initialized vectors (compound or non-compound) to inform the model whether the corresponding instance is a compound or non-compound word. On the other hand, DepNeCTI-XLMR is informed about compound's location in the input string using bracketing (for example, <*sumitrā-ānanda-vardhanaḥ*>) and it lacks span encoding component. Next, to transform the compound-level parsing task into standard dependency parsing, we introduce (1) an additional token called "Global" (▆ color) as a global head for all words in the sentence. (2) The compound head and non-compound words are connected to the Global token using the "Compound Root" and "Global Relation" relations, respectively. The hidden representations of the tokens are obtained using a Bi-LSTM or XLMR encoder. Finally, a Bi-affine (Dozat and Manning, 2017) dependency module is applied on top of the hidden representations.

process creates a nested structure of binary compounds, except for a few exceptions, where the structure represents a constituency tree. However, treating this as a constituency parsing task poses challenges. The nested structure of compounds does not adhere to a syntactic structure but instead follows a semantic structure based on component relations. If we substitute the intermediate nodes with semantic relations, the same spans can be represented as dependency structures by annotating the types as directed relations. Moreover, the headwords in constituency spans are not explicitly marked but can be identified through their corresponding types. In contrast, dependency structures allow the determination of the headword based on labels directed towards it within the compound. Notably, dependency structures faithfully capture

constituency information and can be mutually converted with their corresponding spans (Goyal and Kulkarni, 2014).

Summarily, the semantic relations among compound components resemble dependency relations, which can be represented as directed labels within the dependency parse structure. This approach succinctly represents the semantic relations without introducing intermediary nodes. Lastly, it enables the simultaneous identification of the structure or constituency span alongside the identification of compound types. We encourage readers to refer to Appendix § A for a more detailed illustration.

## 4 DepNeCTI: The Proposed Framework

Figure 4 illustrates the proposed framework with an example "*sumitrā-ānanda-vardhanaḥ lakṣmaṇaḥ*

*rāmam anujagāma*" (Translation: "Lakṣmaṇa, the one who enhanced the delight of Sumitrā, followed Rāma") where '*sumitrā-ānanda-vardhanaḥ*' is a multi-component compound word. We assume prior knowledge of compound segmentation and treat the individual components of multi-component compounds as separate words. We propose two variants of DepNeCTI depending on the choice of encoder: DepNeCTI-LSTM and DepNeCTI-XLMR. In DepNeCTI-LSTM, to differentiate between compound (highlighted with ▇ color) and non-compound (highlighted with ▇ color) tokens, we employ span encoding. The span encoding uses two randomly initialized vectors (compound or non-compound) to inform the model whether the corresponding instance is a compound or non-compound word. On the other hand, DepNeCTI-XLMR is informed about compound's location in the input string using bracketing (for example, *<sumitrā-ānanda-vardhanaḥ>*) and it lacks span encoding component. In order to convert the compound-level dependency parsing task into standard dependency parsing, we introduce two modifications. First, we introduce an additional token called "Global" (▇ color) which serves as the global head for all words in the sentence. Second, we establish connections between the compound head and non-compound words to the Global token using the "Compound Root" and "Global Relation" relations, respectively.

Formally, in a given sentence $X = \{x_1, x_2, ..., x_M\}$ with $M$ tokens, let $x_p$ ($1 \leq p \leq M$) denote an $N$-component compound. Notably, X may contain multiple occurrences of multi-component compounds. The $N$-component compound ($x_p$) is further split into its components ($x_p = \{x_p^1, x_p^2, ..., x_p^N\}$). Next, we pass the overall sequence ($X = \{x_1, x_2, ..., x_p^1, x_p^2, ..., x_p^N ..., x_M\}$) to the encoder to obtain hidden representations. The LSTM encoder concatenates a token's word, character and span embedding to obtain its representation and the XLMR encoder uses word-pieces. Finally, a Bi-affine ([Dozat and Manning, 2017](#)) dependency module is applied on top of it.

## 5 Experiments

### 5.1 Datasets

Table 1 presents data on the total number of multi-component compounds (Figure 5) and their statistics. Our primary focus is on compounds with

more than two components ($n > 2$), while also considering binary compounds if they occur in the context. These datasets comprise segmented compound components, nested spans, context, and semantic relations among the nested spans. We offer two levels of annotations for these datasets: coarse (4 broad types) and fine-grained (86 sub-types). There are 4 broad semantic types of compounds: Avyayībhava (Indeclinable), Bahuvrīhi (Exocentric), Tatpuruṣa (Endocentric) and Dvandva (Copulative). Again, each broader class is divided into multiple subclasss, leading to 86 fine-grained types.[1] Figure 6 shows class-wise label frequency in NecTIS fine-grained.

| Datasets | #Nested | #Train | #Test | #Dev | #Types |
|---|---|---|---|---|---|
| NeCTIS | 17656 | 12431 | 3493 | 2405 | 4 (86) |
| NeCTIS OOD | 1189 | – | 1189 | – | 4 (86) |

Table 1: Data statistics for NeCTIS and NeCTIS-OOD

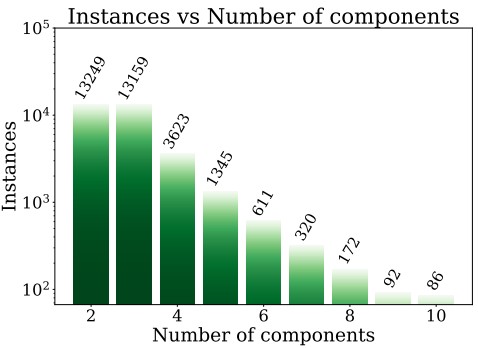

Figure 5: Frequency of $n$-component compounds.

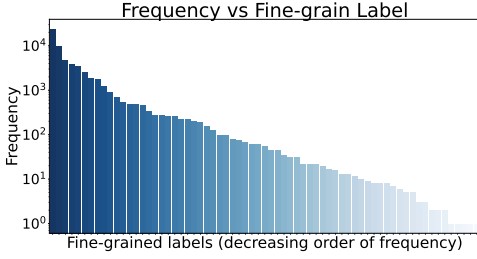

Figure 6: Frequency in NeCTIS fine-grained.

We introduce two context-sensitive datasets: NeCTIS and NeCTIS OOD. The purpose of the additional dataset (NeCTIS OOD) is to create an out-of-domain testbed. The multi-component compound instances are extracted from various books

---

[1]The list of fine-grained labels and the corresponding examples can be found at: https://sanskrit.uohyd.ac.in/scl/GOLD_DATA/Tagging_Guidelines/samaasa_tagging16mar12-modified.pdf

categorized into 4 types based on subject content: philosophical, paurāṇic (Translation: Epic is a genre of ancient Indian literature encompassing historical stories, traditions, and legends.), literary, and āyurvedā. The NeCTIS dataset encompasses compounds from books falling under the Philosophical, Literary, and āyurvedā categories. The multi-component compound instances extracted from the paurāṇic category are included in the NeCTIS out-of-domain (NeCTIS OOD) dataset. Furthermore, the multi-component instances in NeCTIS belong to the prose domain, while NeCTIS-OOD pertains to the poetry domain. Poetry commonly uses multi-component compounding extensively (more exocentric compounds) to adhere to metrical constraints and convey complex concepts. Conversely, prose uses compounds in a more direct and less condensed manner. Furthermore, poets in the realm of poetry often enjoy the freedom to form novel compounds or employ unconventional ones to conform to meter requirements, rendering these compounds infrequent in regular usage.

**Dataset Annotation Process:** We established a data creation process to address the unavailability of annotated context-sensitive multi-component compound data in Sanskrit. We employ a sufficient annotation budget sponsored by DeitY, 2009-2012 for the Sanskrit Hindi Machine Translation project to employ 6 institutes, each consisting of approximately 10 team members. Each team was organized in a hierarchical manner. There were 3 levels in the hierarchy: Junior linguist (Masters degree in Sanskrit), Senior linguist (Ph.D. in Sanskrit) and professional linguist (Professor in Sanskrit). The annotations from lower expertise were further checked as per the above-mentioned hierarchy. Subsequently, the annotated data underwent an exchange process with another team for correctness verification. Any ambiguities encountered during the annotation process were resolved through collective discussions conducted by the correctness-checking team. The available books were distributed among these teams, and each team was responsible for annotating their allocated books. The annotation guidelines[2] are essentially based on Sanskrit grammar which provides the syntactic and semantic criteria for annotation.

Elaborate commentaries accompany the majority

---

[2]The guidelines are available at: https://sanskrit.uohyd.ac.in/scl/GOLD_DATA/Tagging_Guidelines/samaasa_tagging16mar12-modified.pdf

of the texts, that discuss the semantics associated with the compounds, which are typically studied by students as a part of their coursework. Given these considerations, it is very unlikely for professional linguists, often professors instructing these texts, to make mistakes. The dataset was curated around 12 years ago, primarily with the aim of producing error-free gold-standard data. Consequently, the errors made by junior annotators were not recorded or measured, aligning with our focus on achieving error-free quality. The benchmark for determining correctness was based on the Pāṇiṇīan grammar.

## 5.2 Baselines

We investigate the efficacy of various standard formulations (originally proposed for nested named entity recognition for English) for the proposed task. We adapt these systems to the NeCTI task by providing the location of the compounds to ensure a fair comparison with DepNeCTI. Since these baselines are leveraged from the nested named entity recognition task, they do not have explicit channels to provide the location of a compound word. Therefore, we provide this information in the input string itself with the help of brackets (for example, *<sumitrā-ānanda-vardhanaḥ>*):

- **Constituency Parsing (CP):** Following Fu et al. (2020), we formulate NeCTI as constituency parsing with partially-observed trees, with all labeled compound spans as observed nodes in a constituency tree and non-compound spans as latent nodes. We leverage TreeCRF to model the observed and the latent nodes jointly.

- **Bottom-up Constituency Parsing (BotCP):** Following Yang and Tu (2021), we formulate NeCTI as a bottom-up constituency parsing, where a pointer network is leveraged for post-order traversal within a constituency tree to enhance parsing efficiency, enabling linear order parsing.

- **Span Classifier (SpanCL):** Following Yuan et al. (2022), we formulate NeCTI as a span classification problem, where triaffine mechanism is leveraged to learn a better span representation by integrating factors such as inside tokens, boundaries, labels, and related spans.

- **Lexicalized Constituency Parsing (LexCP):** Following Lou et al. (2022), we formulate NeCTI as lexicalized constituency parsing, which embeds a constituency and a dependency trees

| | | Coarse | | | | | | Fine Grain | | | | | |
|---|---|---|---|---|---|---|---|---|---|---|---|---|---|
| | | w/o context | | | w/ context | | | w/o context | | | w/ context | | |
| | Models | USS | LSS | EM | USS | LSS | EM | USS | LSS | EM | USS | LSS | EM |
| **NeCTIS** | BotCP | 72.90 | 58.78 | 32.26 | 76.22 | 63.97 | 35.10 | 74.28 | 33.50 | 18.58 | 75.72 | 41.80 | 23.05 |
| | CP | 76.83 | 61.71 | 39.97 | 64.73 | 46.27 | 30.14 | 77.38 | 41.86 | 27.22 | 70.56 | 32.26 | 21.11 |
| | LexCP | 93.39 | 84.74 | 72.88 | 93.39 | 85.16 | 74.41 | 88.70 | 45.86 | 14.72 | 87.86 | 48.87 | 19.67 |
| | Seq2seq | 92.54 | 84.11 | 59.90 | 91.18 | 80.45 | 52.89 | 92.67 | 65.63 | 30.65 | 92.94 | 68.19 | 34.35 |
| | SpanCL | 92.84 | 84.80 | 69.12 | 93.13 | 84.74 | 69.67 | 93.12 | 69.38 | 52.17 | 92.69 | 68.05 | 50.82 |
| | **DepNeCTI-LSTM** | 95.46 | 89.06 | 76.82 | **97.42** | 89.24 | 77.00 | 95.38 | 79.49 | 57.46 | **97.49** | 79.72 | 56.83 |
| | **DepNeCTI-XLMR** | **96.21** | **90.83** | **79.85** | 96.16 | **90.67** | **79.45** | **96.35** | **83.36** | **63.92** | 96.34 | **83.19** | **63.30** |
| **NeCTIS OOD** | BotCP | 72.81 | 48.08 | 21.63 | 73.89 | 51.18 | 22.57 | 72.30 | 20.52 | 8.64 | 73.87 | 30.10 | 13.32 |
| | CP | 71.43 | 52.65 | 34.57 | 68.17 | 38.63 | 25.15 | 76.12 | 29.68 | 19.77 | 69.57 | 24.10 | 15.43 |
| | LexCP | 89.90 | 71.44 | 50.00 | 91.57 | 72.60 | 52.73 | 83.45 | 32.85 | 7.53 | 83.38 | 34.18 | 8.31 |
| | Seq2Seq | 84.26 | 71.71 | 45.33 | 90.13 | 71.41 | 44.00 | 92.16 | 53.55 | 24.51 | 92.87 | 53.61 | 24.96 |
| | SpanCL | 91.51 | 72.69 | 56.30 | 90.24 | 71.89 | 54.46 | 89.19 | 50.21 | 32.66 | 90.88 | 50.50 | 34.00 |
| | **DepNeCTI-LSTM** | 93.32 | 78.94 | 57.40 | 95.67 | 79.26 | 57.90 | 93.88 | 67.96 | 36.60 | 95.88 | 67.26 | 36.26 |
| | **DepNeCTI-XLMR** | **95.56** | **84.24** | **65.50** | 95.56 | **84.45** | **65.00** | **95.56** | **74.26** | **45.70** | 95.45 | **73.54** | **44.37** |

Table 2: Evaluation on the NeCTIS datasets, considering 2 levels of annotations (coarse and fine-grained) and in 2 settings (with and without context). The best-performing results are in bold, while the second-best results are underlined. The results are averaged over 3 runs. To assess the significance between the proposed system and the best baselines for each setting, a significance test in Accuracy metrics was conducted: $p < 0.01$ (as per t-test).

together. This formulation leverages the constituents' heads into the architecture which is crucial for the NeCTI task.

- **Seq2Seq:** Following Yan et al. (2021), we formulate NeCTI as an entity span sequence generation task using the pretrained seq2seq framework. This generative framework can also identify discontinuous spans; however, NeCTI does not have such instances.

- **DepNeCTI:** We propose two variants of our system (§4) depending on the choice of encoders: DepNeCTI-LSTM and DepNeCTI-XLMR (Nguyen et al., 2021). The compound's location is provided in DepNeCTI-LSTM using span encoding; however, DepNeCTI-XLMR lacks span encoding and leverages this information from the input similar to other baselines.

**Evaluation Metrics:** We evaluate the performance using the Labeled/Unlabeled Span Score (LSS/USS) in terms of micro-averaged F1-score. We define LSS as a micro-averaged F1-score applied on tuples of predicted spans including their labels. We exclude labels of the spans while calculating USS. Additionally, we report the exact match (EM) which indicates the percentage of the compounds for which the predictions of all spans and their semantic relations are correctly identified. Refer to Appendix B for hyper-parameters and details of the computing infrastructure used.

## 5.3 Results

Table 2 presents the performance of the top-performing configurations among all baselines on the NeCTIS benchmark datasets' test set. The evaluation includes 2 levels of annotations (coarse and fine-grained) and in 2 settings (with and without context). While all baseline systems demonstrate competitiveness, no single baseline consistently outperforms the others across all settings. Consequently, we underline the best-performing numbers within each specific setting.

Our proposed system DepNeCTI surpasses all competing systems across all evaluation metrics, demonstrating an absolute average gain of 13.1 points (LSS) and 11.3 points (EM) compared to the best-performing baseline in each setting. Notably, our proposed system exhibits substantial performance superiority over the best baseline in fine-grained settings, particularly in low-resourced scenarios. This validates the effectiveness of our proposed system in low-resourced settings with fine-grained labels. The significant performance gap between our proposed system and the best baselines highlights the efficacy of employing a simple yet effective architecture inspired by the dependency parsing paradigm. These results establish new state-of-the-art benchmarks by integrating the contextual component into our novel framework. While most baselines (except BotCP) do not benefit from contextual information, DepNeCTI-LSTM

demonstrates slight improvements and DepNeCTI-XLMR shows on par improvements when leveraging contextual information. Furthermore, as the number of components grows, the number of potential solutions increases exponentially, leading to poor performance by the systems in such scenarios. Due to this exponential possibility, contextual information provides limited improvements compared to binary compound identification settings (Sandhan et al., 2022a). In other words, unlike the context-free setting, the introduction of context information does not warrant an expectation for the system to precisely generate the correct solution from the exponential candidate space. Figure 3 provides a visual representation that elucidates the concept of this exponential candidate space. A similar performance trend is observed for the NeCTIS-OOD dataset.

# 6 Analysis

Here, we examine the proposed system, focusing on a comprehensive analysis and its applicability. For this purpose, we utilize the NeCTIS coarse dataset under the w/ context configuration. We report LSS in terms of macro-average F1-scores.

**(1) Ablation analysis:** In this study, we analyze the impact of different system components on the overall enhancements of DepNeCTI-LSTM. Ablations, documented in Table 3, present the evaluation metrics when a specific component is deactivated within DepNeCTI-LSTM. For instance, the absence of the span encoding component is denoted as "- Span Encoding". The results indicate that removing any component leads to a decline in performance. Notably, Table 3 highlights the significance of the "Span Encoding" component in driving the improvements.

| System | P | R | F1 | EM |
|---|---|---|---|---|
| DepNeCTI-LSTM | 89.24 | 89.24 | 89.24 | 77.00 |
| - FastText (FT) | 88.84 | 88.84 | 88.84 | 76.50 |
| - Span Encoding (SE) | 86.18 | 85.53 | 86.85 | 70.54 |
| - FT - SE | 84.25 | 84.23 | 84.24 | 67.86 |

Table 3: Each ablation involved the removal of a singular component from DepNeCTI-LSTM. The ablation denoted as "-Span Encoding" entailed eliminating the span encoding component from the proposed system.

**(2) How robust is the system when the number of components of a compound increases?** We analyze the relationship between the F1-score and the number of components in compounds. For compounds with a small number of components, all systems demonstrate high performance, but our proposed systems consistently outperforms other baselines. However, as the number of components increases, the number of examples in each category decreases. Additionally, the number of potential solutions grows exponentially, following the Catalan number. Consequently, all systems experience a decline in performance.

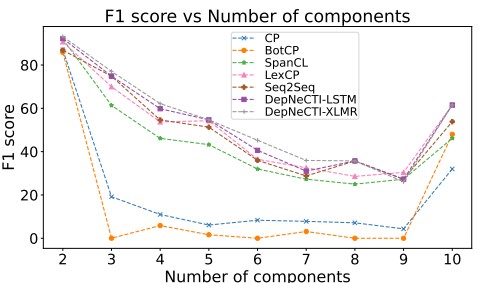

Figure 7: F1-Score against the number of components. The compounds with components $N > 10$ are excluded.

**(3) Error analysis:** We investigate whether all the systems are able to identify the location of a multi-component compound correctly. The motivation behind this experiment is to evaluate the capability of the baselines and the proposed architecture to leverage the information about the compound's location effectively. We define the span of text that corresponds to a compound as a global span which we know apriori. Figure 8 illustrates the effectiveness of each system in correctly identifying the global span of multi-component compounds. In

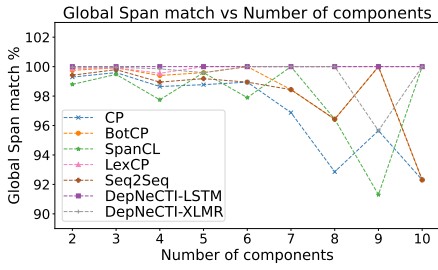

Figure 8: Performance of the systems on identifying the global span of a compound.

DepNeCTI-LSTM, our span encoding effectively captures this information, resulting in a perfect 100% score. However, even after providing the baselines with this information, they fail to use it due to limitations in their architectures. Interestingly, DepNeCTI-XLMR does not contain a span

encoding component and leverages the compound's location information from the input as provided for the baselines. Still, DepNeCTI-XLMR reports the best performance due to its powerful word representation ability. It is worth noting that the NeCTIS dataset exhibits an inherent bias towards left branching, as indicated by the nested tree structure. Consequently, all systems display a bias towards left branching as well. Therefore, due to the dominance of left-branching instances and increased variance due to less number of instances, a spike is observed in the results.

**(4) Efficiency of our proposed system:** Figure 9 present the computational efficiency of our system measured in terms of the number of sentences processed per second. We compare the inference speed of different baselines on the NeCTI task. Our systems, DepNeCTI-LSTM and DepNeCTI-XLMR are leveraging a simple architecture and utilizing dependency parsing as an appropriate problem formulation, exhibits a 5-fold/3-fold improvement over the most efficient baseline, BotCP.

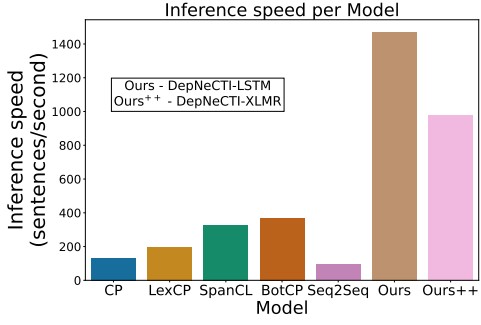

Figure 9: Inference speed of the competing systems.

## 7 Related Works

**Lexical semantics** is a dedicated field focused on word meaning. Various tasks such as word-sense disambiguation (Bevilacqua and Navigli, 2020; Barba et al., 2021; Maurya et al., 2023; Maurya and Bahadur, 2022), relationship extraction (Tian et al., 2021; Nadgeri et al., 2021; Hu et al., 2022; Wang et al., 2022), and semantic role labeling (He et al., 2018; Kulkarni and Sharma, 2019; Zhang et al., 2022) play essential roles in determining word meaning. Moreover, when dealing with complex word structures such as compounds, named entities, and multi-word expressions, relying solely on basic word senses and relationships is inadequate. While efforts have been made in Noun Compound Identification (Ziering and van der Plas,

2015; Dima and Hinrichs, 2015; Fares et al., 2018; Shwartz and Waterson, 2018; Ponkiya et al., 2018, 2020, 2021), multi-word expression (MWE) (Constant et al., 2017; Gharbieh et al., 2017; Gooding et al., 2020; Premasiri and Ranasinghe, 2022) and named entity recognition (Fu et al., 2020; Yang and Tu, 2021; Lou et al., 2022; Yuan et al., 2022), the nested compounds remains unexplored.

**Sanskrit Compound Type Identification** has attracted significant attention over the past decade. Decoding the meaning of a Sanskrit compound requires determining its constituents (Huet, 2009; Mittal, 2010; Hellwig and Nehrdich, 2018), understanding how these constituents are grouped (Kulkarni and Kumar, 2011), identifying the semantic relationship between them (Kumar, 2012), and ultimately generating a paraphrase of the compound (Kumar et al., 2009). Previous studies proposed rule-based approaches (Satuluri and Kulkarni, 2013; Kulkarni and Kumar, 2013), a data-driven approach (Sandhan et al., 2019) and a hybrid approach (Krishna et al., 2016) for SaCTI. Sandhan et al. (2022a) proposed a context-sensitive architecture for binary compounds.

Earlier works in Sanskrit solely focused on binary compounds, neglecting the identification of multi-component compound types; however, our proposed framework fills this research gap.

## 8 Conclusion

In this work, we focused on multi-component compounding in Sanskrit, which helps to decode the implicit structure of a compound to decipher its meaning. While previous approaches have primarily focused on binary compounds, we introduced a novel task called nested compound type identification (NeCTI). This task aims to identify nested spans within multi-component compounds and decode the implicit semantic relations between them, filling a gap in the field of lexical semantics. To facilitate research in this area, we created 2 newly annotated datasets, designed explicitly for the NeCTI task. These datasets were utilized to benchmark various problem formulations. Our novel framework DepNeCTI outperformed the best baseline system by achieving a stupendous absolute gain of 13.1 points F1-score in terms of LSS. Similar to the previous findings on binary Sanskrit compound identification, we discovered that our proposed system exhibits substantial performance superiority over the best baseline in low-resourced scenarios.

## Limitations

We could not extend our framework to other languages exhibiting multi-component compounding phenomena due to the lack of availability of annotated datasets. It would be interesting to measure the effectiveness of rules from Pāṇinīan grammar to discard incompatible semantic relations (Kulkarni, 2019, 2021).

## Ethics Statement

This work introduces a new task, along with annotated datasets and a framework, to address the nested compounding phenomenon in Sanskrit. The proposed resources aim to enhance the understanding of multi-component compounds and contribute to the improvement of machine translation systems. Regarding potential effects, we anticipate no harm to any community resulting from the use of our datasets and framework. However, we advise users to exercise caution, as our system is not flawless and may generate mispredictions. To ensure transparency and future research, we have publicly released all our annotated NeCTIS datasets and source codes. We confirm that our data collection adheres to the terms of use of the sources and respects the intellectual property and privacy rights of the original authors. Our annotation team consisted of qualified individuals, including Master's and Ph.D. degree holders, some of whom are Sanskrit professors. Annotators were compensated appropriately and provided with detailed instructions to ensure consistency in the annotation process. We remain committed to addressing ethical implications as we refine our systems and welcome feedback from the community to enhance our ethical practices.

## Acknowledgements

We thank all the annotators from different institutes for helping us with NeCTIS data annotation. We would like to thank Zheng Yuan, Tsinghua University (Alibaba Group), for helping us with the hyperparameter section and adapting his system for the NeCTI task. We appreciate and thank all the anonymous reviewers for their constructive feedback towards improving this work. The work was supported in part by the National Language Translation Mission (NLTM): Bhashini project by Government of India.

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

## A  Why NeCTI as a Dependency Parsing Task?

Compounds in language are semantic constructions. While a limited number of rules derived from Sanskrit Grammar aid in determining the syntactic structure of a compound, they offer limited assistance in uncovering its meaning. The meaning of a compound primarily arises from the semantic relationship between its components. This semantic relationship, known as "sāmarthya," is expressed through various types of semantic compounds. These compound types also facilitate the identification of the headword within a compound. The headword can be one of the constituents or an entirely distinct word modified by the resulting compound. Compounds consisting of more than two components are typically formed through successive binary combinations of the components, following specific relations. As a result, a nested structure of binary compounds is created, with a few exceptions. Therefore, the semantic compound types play a vital role in determining the correct nested structure amidst the various possible structures of a compound. Each of these structures represents a constituency span, and a parser equipped with compound type identification assists in accurately identifying the appropriate constituency span.

Treating this as a constituency parsing task presents several challenges. Figure 10 illustrates the potential constituency spans for a 3-component compound $a - b - c$, with $ab$ and $bc$ representing the intermediate compounds. First, the nested structure of compounds does not conform to a syntactic structure; instead, it follows a semantic structure based on the relations between the components rather than their position or relative co-occurrence. Second, the intermediate compounds within the structure are not categories but stem forms. Each intermediate compound serves as an entity that modifies the meanings of one of its constituents, subsequently combining with another component to form a larger compound.

Kumar (2012) formulated this as constituency parsing, where the identification of semantic compatibility was performed based on the relative co-occurrence and relative position of the components. As shown in Figure 11, the intermediate compounds, which lacked additional information, were substituted with their respective semantic compound types in the type identification stage.

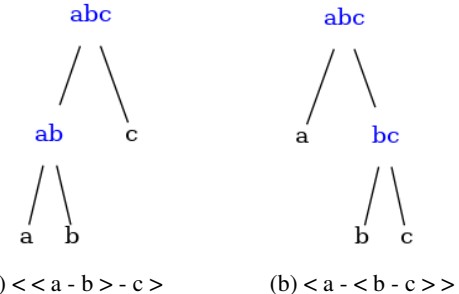

(a) < < a - b > - c >          (b) < a - < b - c > >

Figure 10: Possible constituency spans for a three component compound a-b-c

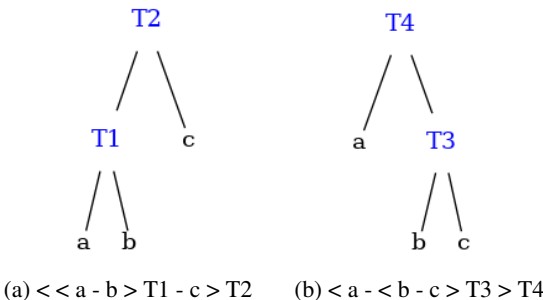

(a) < < a - b > T1 - c > T2     (b) < a - < b - c > T3 > T4

Figure 11: Possible constituency spans for a three component compound a-b-c with semantic types

The semantic types serve as semantic constructions for compounds, rather than being syntactic categories and play a crucial role in determining the meaning of the compound. For example the compound *vidyālayaghaṇṭā* has the nested structure «*vidyā-ālaya>T6-ghaṇṭā>T6*. The type T6 indicates *ṣaṣṭhī tatpuruṣa* compound ($6^{th}$ case determinative compound) inferring a possessive relationship. *Vidyā* (knowledge) and *ālaya* (place) combine to form the intermediate compound *vidyālaya* (the place of knowledge, viz. school), which combines with *ghaṇṭā* (bell) to form the whole compound indicating school-bell. The possessive relation expressed by the compound type (T6) is akin to a dependency relation, and this holds true for other compound types as well.

Consequently, when represented with compound types, the same constituency spans can be depicted as dependency structures by annotating the types as directed relations and removing the extra nodes indicating the types. The dependency structures for the previous example ($a - b - c$) are shown in Figure 12. Furthermore, the head words are not explicitly marked in constituency spans and can only be identified through their corresponding types. However, with dependency structures, the head word can be determined by the labels directed towards it within the compound. Notably, these dependency structures faithfully capture the constituency information and are mutually convertible with their corresponding constituency spans.

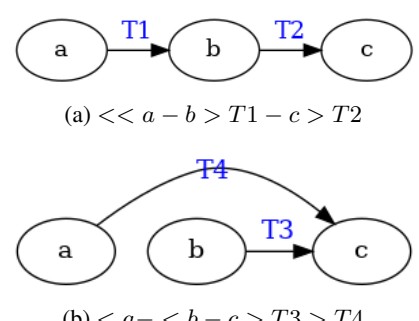

(a) < < a − b > T1 − c > T2

(b) < a− < b − c > T3 > T4

Figure 12: Dependency Structure for the three component compound a-b-c with semantic types as labels

There are several considerations behind the decision to treat NeCTI as a dependency parsing task. First, the semantic relations among compound components resemble dependency relations and can be represented as directed labels within the dependency parse structure. Second, this approach concisely represents the semantic relations between compound components without introducing extra intermediary nodes. Lastly, it enables the simultaneous identification of the structure or constituency span alongside the identification of compound types.

## B  Experiment Details

**Hyper-parameters:**  For our proposed system, we build on the top of the codebase from BiAFF, as developed by Ma et al. (2018). We configure the hyperparameters as follows: a batch size of 16, 100 training iterations, a dropout rate of 0.33, 2 stacked Bi-LSTM layers, a learning rate of 0.002, and the remaining parameters set identically to those used in the work of Ma et al. (2018). We use manual tuning for the hyper-parameter selection and F1-score criteria on dev set's performance . Our codebase is publicly available and released under a creative-common license. We use FastText word embeddings for the proposed framework.

**Computing Infrastructure Used:**  We perform our experiments using a single GPU equipped with an NVIDIA A40, 48 GB GPU memory, and 10752 CUDA cores. We employ a single GPU with an NVIDIA Quadro RTX 4000, 8 GB GPU memory, and 2304 CUDA cores for our proposed system.