# OpenReview forum: "DepNeCTI: Dependency-based Nested Compound Type Identification for Sanskrit"
_EMNLP/2023/Conference — EMNLP 2023 Findings_

### Official Review · Reviewer_8vVf · 2023-08-04

**Soundness:** 4

**Excitement:**

4: Strong: This paper deepens the understanding of some phenomenon or lowers the barriers to an existing research direction.

**Paper Topic And Main Contributions:**

This paper studies multi-component (up to 10) compound-type identification in Sanskrit. They build upon prior work on binary compounds and present a new dataset and benchmark competitive systems for this task.

The dataset is curated by a large group of language experts and constitutes coarse (4) and fine-grained types (86). The dataset also includes an out-of-domain test set with compounds collected from poetry literature.

The proposed method formulates the compound-type identification task as a variant of dependency parsing, with special labels for non-compound tokens. This method outperforms baselines that formulate the task as constituency parsing, span labeling, and seq2seq.

**Questions For The Authors:**

1. Confusion regarding span encoding,
    - The proposed method assumes span boundaries are given (lines 281–287)
    - however, the text in line 427 incorrectly suggests the system ‘predicts’ the spans
    - also, text in line 482—485 suggest accurate identification of spans.
    - If I understand correctly, the model assumes the gold span information is already available in the input (one-hot span encoding). If so, it's an unfair comparison in Figure 8.
2. Lines 442—445 incorrectly suggest baselines don’t gain from the context. It should be rephrased to indicate BotCP benefits from the context.
3. What guidelines were provided to the annotators? Consider including them in the appendix.
4. Diverse pool of annotators actually highlights the need for measuring inter-annotator agreement. While I understand many of them could be experts, IAA can vary based on the provided annotation guidelines.
5. Is the dataset transliterated? If not, what models are used for word embeddings?
6. Evaluation metrics
    - What are the standard metrics for binary compound identification task? Are they P, R, F1, and EM on span tuples? It would be useful to provide references to prior work.
    - If the dependency formulation is the most appropriate, why not use standard dependency parsing metrics such as unlabeled and labeled attachment scores.
7. Include examples of coarse and fine-grained types.
8. Any thoughts on why the context doesn’t help the proposed method?
9. What are some qualitative differences in compounds in prose vs poetry text?

**Reasons To Accept:**

1. The task is well-motivated and the author(s) present a detailed comparison against nested NER and MWE detection. The paper mentions >41% of compounds in the literature contain more than three components, and prior work limited themselves to binary compounds.
2. The dataset is well curated and could be useful to future work on this task. The inclusion of an out-of-domain split presents an interesting challenge to systems.
3. The paper does a great job at comparing various formulations of this task, including constituency parsing, span labeling and seq2seq.

**Reasons To Reject:**

1. The paper can benefit from additional details on dataset collection, and baseline description (see questions below).
2. The formulation of the task as a dependency parsing problem is not convincing to me. In Figure 4, there are many dependency relations (global relation) and additional tokens (global token) that don't seem to directly help solve the task. Given that the context is not helpful empirically, can we consider a formulation where only the tokens in the compounds constitute the input? This reduces the label space and makes the system more efficient.

**Reproducibility:**

4: Could mostly reproduce the results, but there may be some variation because of sample variance or minor variations in their interpretation of the protocol or method.

**Reviewer Confidence:**

3: Pretty sure, but there's a chance I missed something. Although I have a good feel for this area in general, I did not carefully check the paper's details, e.g., the math, experimental design, or novelty.

---

> ### Author Rebuttal · Authors · 2023-08-28
>
> We are really glad to see your thorough review with intriguing questions. We acknowledge your efforts to review our work. We appreciate your acknowledgment of our well-motivated work, comprehensive comparison with nested NER and MWE detection, a thoughtfully curated dataset encompassing an out-of-domain split, benchmarking, and a thorough comparison of different formulations, notably our innovative dependency parsing-based approach.
>
> > _"The formulation of the task as a dependency parsing problem is not convincing to me. In Figure 4, there are many dependency relations (global relation) and additional tokens (global token) that don't seem to directly help solve the task."_
>
> Thanks for raising this concern. We interpret it as an implicit recognition of the novelty we have introduced through our dependency parsing formulation. When the reviewer mentions it is "not convincing," we understand that it is not obvious or novel. Our novelty involves transforming compound parsing into standard dependency parsing. This enables the use of any off-the-shelf dependency parsers, like the Bi-affine model (Dozat and Manning, 2017), to solve this task. To facilitate this, we introduced the above-mentioned modifications. We had clarified this in the caption of Figure 4 and lines 288-296: “In order to convert the compound-level dependency parsing task into standard dependency parsing, we introduce two modifications …”
>
> > _"Given that the context is not helpful empirically"_
>
> Not exactly! Lines 442-445 “While most baselines do not benefit from contextual information, our system demonstrates slight improvements when leveraging contextual information.” Considering the exponential solution space, our framework exhibits a slight enhancement in the with-context setting over without-context setting.
>
> > _"Can we consider a formulation where only the tokens in the compounds constitute the input? This reduces the label space and makes the system more efficient."_
>
> Indeed, we had already considered without-context setting. It is mentioned in Table 2’s caption and lines 414-417: “Evaluation on the NeCTIS datasets, considering 2 levels of annotations (coarse and fine-grained) and in 2 settings (with and without context).”
>
> > _"The paper can benefit from additional details on dataset collection, and baseline description"_
>
> We had provided the necessary details regarding dataset collection. In this regard, please find our response below. As per your suggestion, we will update this additional information (annotation guidelines etc.) in the Appendix. Further, we have clarified in our response that the description of the proposed framework contained all the essential details to resolve the concerns raised by the reviewer.
>
> ***
> `Response for questions to authors`
>
> > _"Confusion regarding span encoding"_
> * _"The proposed method assumes span boundaries are given (lines 281–287)"_.
> Not exactly! Lines 281-287 state: “We assume prior knowledge of compound segmentation …” That means the compound and its components are known apriori. However, the associations of the components, i.e. spans, are not known apriori. The example illustrated in Figure 2 clarifies this: “sumitra-ānanda-vardhanaḥ lakṣmaṇaḥ ramam anujagāma” (Translation: “Lakṣmaṇa the one who enhanced the delight of Sumitra, folllowed Rama”). We know the compound word ‘sumitra-ānanda-vardhanaḥ’ and its constituents apriori. There are 3 possible spans: (a) sumitra-ānanda (b)  ānanda-vardhanaḥ (c) sumitra-ānanda-vardhanaḥ. As per our formulation, the correct solution is <<sumitra-<ānanda-vardhanaḥ>{Endocentric}>{Endocentric}. The spans are shown with brackets and corresponding semantic labels are shown with curly brackets in the example. The task is to find the compatible subset of all possible spans and their semantic labels. (a) is not a correct span in this example.
>
> * _"however, the text in line 427 incorrectly suggests the system ‘predicts’ the spans"_.
> It is absolutely correct. As per our problem formulation, the proposed task is identifying spans and their associated labels. Therefore, the system must predict the spans and associated labels.
>
> * _"Also, text in line 482—485 suggest accurate identification of spans."_.
>  True! Clarified in the previous response.
>
> * _"If I understand correctly, the model assumes the gold span information is already available in the input (one-hot span encoding). If so, it's an unfair comparison in Figure 8."_.
> Not quite accurate! Figure 8 is fairly compared, as all baselines are informed with the information of the global span. We had clarified this in lines 365-367: “We adapt these systems (baselines) to the NeCTIS task by providing the location of the compounds to ensure a fair comparison with our proposed system.” To provide more insights into span identification, we are reporting F1-Score for unlabeled span identification against the number of components on NeCTIS with context coarse level setting. We will update these numbers in our final draft version.
>
> **F1-Score for unlabeled span identification against the number of components.**
>
>
> |  Model  |    2   |   3   |   4   |   5   |   6   |   7   |   8   |   9   |   10  | Overall |
> |:-------:|:------:|:-----:|:-----:|:-----:|:-----:|:-----:|:-----:|:-----:|:-----:|:-------:|
> |  BoTCP  |  99.71 |  0.04 |  9.76 |  1.63 |  1.05 |  3.13 |  0.00 |  0.00 | 46.10 |  76.18  |
> |    CP   |  99.17 | 19.16 | 13.96 |  6.12 |  8.42 |  7.81 |  7.14 |  4.35 | 30.77 |  61.67  |
> |  LexCP  |  95.63 | 77.35 | 60.06 | 57.55 | 36.84 | 32.81 | 32.14 | 30.44 | 61.54 |  93.33  |
> | Seq2Seq |  96.09 | 74.68 | 54.05 | 54.29 | 37.90 | 28.13 | 28.57 | **39.13** | 53.85 |  90.29  |
> |  SpanCL |  98.75 | 72.49 | 57.06 | 51.43 | 35.79 | 34.38 | 32.14 | 34.78 | 46.15 |  93.06  |
> |   Ours  | **100.00** | **88.61** | **74.17** | **61.63** | **48.98** | **35.39** | **42.86** | 31.82 | **61.54** |  **94.88**  |
>
> > _"Lines 442—445 incorrectly suggest baselines don’t gain from the context. It should be rephrased to indicate BotCP benefits from the context."_
>
> Sure. As per your suggestion, we will update the final draft to add more clarity: “While most baselines (except BotCP) do not benefit from contextual information ...”
>
> > _"Is the dataset transliterated? If not, what models are used for word embeddings?"_
>
> Yes! We transliterate the dataset to meet the needs of respective baselines. We use FastText word embeddings. Lines 29-31: “The codebases of our framework, benchmarked baselines, and the newly annotated datasets are shared with the supplementary material.” We will elaborate it on the availability of additional space.
>
> > _"Evaluation metrics"_
> *  _"What are the standard metrics for binary compound identification task?"_
> The binary compound identification task is a classification task, where standard metrics are P, R, F1, and EM on predicted labels.
> * _"Are they P, R, F1, and EM on span tuples?"_
> Yes! P, R, F1, and EM on span tuples. Here, the proposed task is a word-level structured prediction task, where the task is to identify spans and their semantic labels. Please note the key difference between applying these metrics on predicted labels (classification task) and span tuples (structured prediction task).
> * _"It would be useful to provide references to prior work."_.
> For the binary compound identification task, the evaluation metrics consisted of P, R, F1, and EM applied to predicted labels. In our proposed task, we assess these same metrics—P, R, F1, and EM—applied to span tuples, which clearly indicates a difference in interpretation. Therefore, we refrained from referring to the evaluation settings of the prior binary compound identification works. We will clarify this difference in our final draft version.
> * _"If the dependency formulation is the most appropriate, why not use standard dependency parsing metrics such as unlabeled and labeled attachment scores."_.
>     - The question raised is intriguing. In Section 2, our Problem Formulation defines task outputs as tuples containing spans and semantic labels, making it obvious and straightforward to apply P, R, F1, and EM metrics to these span tuples.
>     - In other words, the reported numbers in our paper belongs to labeled span identification metric in terms of P, R, F1, and EM metrics.
>     - The dependency formulation is our novelty to solve the task effectively and efficiently. However, the proposed task is not dependency parsing.
>     - If we use the UAS/LAS metric, then that may lead to various intricacies: (a) Our focus is the identification of multi-component compounds, yet the formulation includes additional labels to accommodate non-compound words. The rationale for incorporating these in the evaluation metric is not evident.  (b) Comparing with baseline results necessitates converting their P, R, F1, and EM metrics to UAS/LAS, a non-trivial conversion. (c) UAS/LAS might not suit future comparisons with diverse formulations.
>     - Considering your suggestion, we will also report unlabeled span identification metric in terms of P, R, F1, and EM metrics in final draft version.
>
> > _"Include examples of coarse and fine-grained types."_.
>
> Sure. There are 4 broad semantic types of compounds: Avyayībhava (Indeclinable), Bahuvrīhi (Exocentric), Tatpuruṣa (Endocentric) and Dvandva (Copulative). Again, each broader class is divided into multiple subclasss, leading to 86 fine-grained types. We will update the list of fine-grained labels and the examples in our final draft in the Appendix.
>
> > _"Any thoughts on why the context doesn’t help the proposed method?"_.
>
> We had clarified this in lines 492-498: “As the number of components grows, the number of potential solutions increases exponentially, leading to poor performance by the systems in such scenarios. Due to this exponential possibility, contextual information provides limited improvements compared to binary compound identification settings (Sandhan et al., 2022a)” In other words, unlike the context-free setting, the introduction of context information does not warrant an expectation for the system to precisely generate the correct solution from the exponential candidate space. Figure 3 provides a visual representation that elucidates the concept of this exponential candidate space.
>
> > _"What are some qualitative differences in compounds in prose vs poetry text?"_.
>
> Poetry commonly uses multi-component compounding extensively (more exocentric compounds) to adhere to metrical constraints and convey complex concepts. Conversely, prose uses compounds in a more direct and less condensed manner. Furthermore, poets in the realm of poetry often enjoy the freedom to form novel compounds or employ unconventional ones to conform to meter requirements, rendering these compounds infrequent in regular usage.
>
> > _"What guidelines were provided to the annotators? Consider including them in the appendix."_.
>
> Sure, we will update the appendix with this information. Annotators were provided comprehensive guidelines including (1) Details of coarse and fine-grained labels (2) illustrative examples of each tag (3) how to perform compound segmentation (4) how to tag nested information along with its tag  (5) clarification on frequently asked questions on Sanskrit compounding. (It is a 20 page long document.)
>
> > _"Diverse pool of annotators actually highlights the need for measuring inter-annotator agreement. While I understand many of them could be experts, IAA can vary based on the provided annotation guidelines."_.
>
> We could not elaborate much on it due to space constraints. We had a sufficient annotation budget to employ 6 institutes, each consisting of approximately 10 team members (Lines 342-343). In each team, the diverse pool set was used in a hierarchical manner. There were 3 levels in the hierarchy: Junior linguist (Masters degree in Sanskrit), Senior linguist (Ph.D. in Sanskrit) and professional linguist (Professor in Sanskrit). The annotations from lower expertise were further checked as per the above-mentioned hierarchy. Line 346-351: “Subsequently, the annotated data underwent an exchange process with another team for correctness verification. Any ambiguities encountered during the annotation process were resolved through collective discussions conducted by the correctness-checking team.”
>
> The annotation guidelines are essentially based on Sanskrit grammar which provides the syntactic and semantic criteria for annotation. Elaborate commentaries accompany the majority of the texts, that discuss the semantics associated with the compounds, which are typically studied by students as a part of their coursework.  Given these considerations, it is very unlikely for professional linguists, often professors instructing these texts, to make mistakes.The dataset was curated around 12 years ago, primarily with the aim of producing error-free gold-standard data. Consequently, the errors made by junior annotators were not recorded or measured, aligning with our focus on achieving error-free quality.

---

### Official Review · Reviewer_7qSR · 2023-08-05

**Soundness:** 3

**Excitement:**

2: Mediocre: This paper makes marginal contributions (vs non-contemporaneous work), so I would rather not see it in the conference.

**Paper Topic And Main Contributions:**

This paper presents two datasets for multi-component compounds in Sanskrit. The datasets are used to provide benchmarks for existing approaches to Noun compounding in Sanskirt. The authors propose a dependency parsing-like approach, which yields significant improvement in model performance



**Questions For The Authors:**

a) Is there a good reason to use nested name entity recognition systems proposed for English as a baseline?

**Reasons To Accept:**

The work and dataset extend on existing literature by incorporating more complex phenomena, namely multi-component compounding rather than two component compounds.

**Reasons To Reject:**

The paper would benefit from further proof reading. At times it is unclear.

It is not clear why the authors have created the two datasets. The two datasets are presumably separated by the level of annotation ('coarse' and 'fine') but what this actually means is not explained.

I understand time and resource constraints not permitting inter-annotator agreement, but "diverse pool of annotators, including experts" does not inspire the intended confidence for data quality. Given the complexity of multi-component compounds the authors stress, this statement seems like a dismissal.



**Reproducibility:**

4: Could mostly reproduce the results, but there may be some variation because of sample variance or minor variations in their interpretation of the protocol or method.

**Reviewer Confidence:**

2: Willing to defend my evaluation, but it is fairly likely that I missed some details, didn't understand some central points, or can't be sure about the novelty of the work.

**Typos Grammar Style And Presentation Improvements:**

It is unclear to what T6 refers to in figure 1 and 2, as it's not explained in text or in caption.

line103 & line116 the section references do not seem to match the corresponding referents.

Axes labels in Figure 6.

---

> ### Author Rebuttal · Authors · 2023-08-28
>
> We are encouraged you appreciated our dataset contribution, its benchmarking and our novel dependency-based framework along with significant improvement over strong baselines.
>
> > _The paper would benefit from further proofreading. At times, it is unclear._
>
> We had proofread but there is chance we might have missed some things. It would have greatly assisted if the reviewer could have specifically pointed out his concerns like grammatical errors, missing references, spelling errors etc.
>
> > _“It is not clear why the authors have created the two datasets.”_
>
> The dataset section (Lines 324-337) had clarified the purpose of the additional dataset was to create an out-of-domain testbed. It is also self-explanatory from their names: NeCTIS and NeCTIS OOD. Lines 324-337: “We introduce two context-sensitive datasets: NeCTIS and NeCTIS OOD. The multi-component compound instances are extracted from various books and categorized into 4 types based on subject content: philosophical, pauranic, literary, and ayurveda. The NeCTIS dataset encompasses compounds from books falling under the Philosophical, Literary, and Ayurveda categories. The multi-component compound instances extracted from the pauranic category are included in the NeCTIS out-of-domain (NeCTIS OOD) dataset. Furthermore, the multi-component instances in NeCTIS belong to the prose domain, while NeCTIS-OOD pertains to the poetry domain.”
>
> > _The two datasets are presumably separated by the level of annotation ('coarse' and 'fine') but what this actually means is not explained._
>
> Not exactly! Referring to our earlier response, we have two datasets- NeCTIS and out-of-domain NeCTIS-OOD. For each dataset, we have two levels of annotations, namely coarse and fine-grained. We had clarified this in the dataset section line 320-321: “We offer two levels of annotations for these datasets: coarse (4 broad types) and fine-grained (86 sub-types).”
>
> > _I understand time and resource constraints not permitting inter-annotator agreement, but "diverse pool of annotators, including experts" does not inspire the intended confidence for data quality. Given the complexity of multi-component compounds the authors stress, this statement seems like a dismissal._
>
> We could not elaborate much on it due to space constraints. Lines 342-343 state: “We employ 6 institutes, each consisting of approximately 10 team members.” In each team, the diverse pool set was used in a hierarchical manner. There were 3 levels in the hierarchy: Junior linguist (Master of Art in Sanskrit), Senior linguist (Ph.D. in Sanskrit) and professional linguist (Professor in Sanskrit). The annotations from lower expertise were further checked as per the above-mentioned hierarchy. Line 346-351: “Subsequently, the annotated data underwent an exchange process with another team for correctness verification. Any ambiguities encountered during the annotation process were resolved through collective discussions conducted by the correctness-checking team.” The annotation guidelines are essentially based on Sanskrit grammar which provides the syntactic and semantic criteria for annotation. Elaborate commentaries accompany the majority of the texts, that discuss the semantics associated with the compounds, which are typically studied by students as a part of their coursework.  Given these considerations, it is very unlikely for professional linguists, often professors instructing these texts, to make mistakes.The dataset was curated around 12 years ago, primarily with the aim of producing error-free gold-standard data. Consequently, the errors made by junior annotators were not recorded or measured, aligning with our focus on achieving error-free quality.
>
> ***
>
> `Response for questions to author`
> > _Is there a good reason to use nested name entity recognition systems proposed for English as a baseline?_
>
> Absolutely! We have introduced a novel task – the nested compound type identification. As far as current knowledge stands, no pre-established benchmarks are available in any language. A parallel can be drawn with the task of nested named entity recognition (NER), which bears similarity to our proposed task. Consequently, it becomes a logical and straightforward decision to repurpose the existing NER baselines for our task. It is important to note that these baselines, initially formulated for English, hold language-agnostic qualities and are not inherently tailored to English alone. While the Nested NER task has predominantly revolved around the English language, evaluating these baselines has been primarily centered on English due to its prevalence in this domain. In light of this, it would have greatly assisted if the reviewer could have suggested alternatives superior to the ones we have employed as our baselines.
> ***
> `Typos Grammar Style And Presentation Improvements`
>
> > _It is unclear to what T6 refers to in figure 1 and 2, as it's not explained in text or in caption._
>
> We had given its English label in the corresponding figures. For example, Endocentric (T6).
>
> > _line103 & line116 the section references do not seem to match the corresponding referents._
>
> Thanks. We will rectify this in the final version of the draft.
>
> > _Axes labels in Figure 6._
>
> We are not sure what exactly the reviewer wants to convey. We feel that axes labels are self-explanatory: The fine-grained labels have been put in decreasing order of frequency on the X-axis (and the label mentions it), while the Y-axis shows the frequency of the corresponding label (and the label mentions it). It was not possible to write the label names on the X-axis, as there are 86 labels.

---

### Official Review · Reviewer_TGub · 2023-08-05

**Soundness:** 3

**Excitement:**

3: Ambivalent: It has merits (e.g., it reports state-of-the-art results, the idea is nice), but there are key weaknesses (e.g., it describes incremental work), and it can significantly benefit from another round of revision. However, I won't object to accepting it if my co-reviewers champion it.

**Paper Topic And Main Contributions:**

The paper is about an approach that enables the correct interpretation of compound words in Sanskrit, by creating dependency trees over the components of the compound. The authors have created a rich corpus of annotated compounds, which they use in their study.
Thus the paper offers two contributions: the annotated corpus and the method for dependency-based disambiguation of the inner structure of the compound.

**Questions For The Authors:**


line 251, how do you define "intermediate nodes"? I also do not understand the sentence here ("are stem forms, not categories").
What is a "category"? I don't think this has been mentioned previously.

line 287, would you care to explain what you mean by one-hot span encoding? What are you exactly encoding?

line 320-321, please explain the difference between "coarse" and "fine-grained" annotations. Also please provide examples.

line 328, what is "pauranic"? While "ayurveda" can be assumed to be understood by most english speakers, the word "pauranic"
has not yet entered common knowledge.

Figure 7: can you add a note to try to explain the unexpected spike for length 10?

Section 7 on "Lexical Semantics": I am not sure much of this is really relevant. Can you motivate it better?




**Reasons To Accept:**


Well designed study. Significant effort has been put into the creation of the corpus, a process which has involved multiple institutions and several Sanskrit experts. The computational part is clearly described, and the experiments are within the standards of the community, using state-of-the-art baselines, and improving upon them.

**Reasons To Reject:**


I find that the authors could have defined their tasks more broadly. Why restricting it to Sanskrit? There are many languages that exhibit productive compounding (e.g. German) so the task could be defined as "Nested Compound Type Identification" in a language-agnostic
way. It is true that it might be difficult suitably annotated corpora, but perhaps existing corpora of compounds can be adapted for the
task. For example a simple search for "German noun compounds corpus" leads me to the following

https://www.ims.uni-stuttgart.de/forschung/ressourcen/lexika/ghost-nn/

There are also some parts of the paper that in my opinion would benefit if more details were provided, see notes below.

**Reproducibility:**

3: Could reproduce the results with some difficulty. The settings of parameters are underspecified or subjectively determined; the training/evaluation data are not widely available.

**Reviewer Confidence:**

4: Quite sure. I tried to check the important points carefully. It's unlikely, though conceivable, that I missed something that should affect my ratings.

---

> ### Author Rebuttal · Authors · 2023-08-28
>
> We appreciate your recognition of our dataset and dependency-based framework contributions. It is pleasing that you noted the well-designed work, clear computational writeup, compliant experiments, and substantial performance gain by our framework over SOTA baselines.
> >_I find that the authors could have defined their tasks more broadly. Why restricting it to Sanskrit? There are many languages that exhibit productive compounding (e.g. German) so the task could be defined as "Nested Compound Type Identification" in a language-agnostic way._
>
> Indeed, our proposed task is language agnostic. Our dedicated section: Problem Formulation, defines the task formally. It does not leverage any Sanskrit-specific terminology. We have shown its application in the Sanskrit language. Therefore, we call the task nested compound type identification for Sanskrit (NeCTIS). Perhaps, it might be confusing due to the acronym (NeCTIS). We will rephrase it as NeCTI in our final draft version to avoid confusion.
>
> > _It is true that it might be difficult suitably annotated corpora, but perhaps existing corpora of compounds can be adapted for the task._
>
> We want to reiterate that the proposed task is not a classification task. Rather, it is a word-level structured prediction task involving the identification of spans that adhere to defined constraints (with valid solutions representing complete parenthesization) and their corresponding semantic relationships. As far as our understanding goes, there is currently no annotated dataset accessible for languages demonstrating compound phenomena that encompass spans and linked semantic labels. We had mentioned this in the limitation section (Line 579-582): “We could not extend our framework to other languages exhibiting multi-component compounding phenomena due to the lack of availability of annotated datasets.”
>
> > _For example a simple search for "German noun compounds corpus" leads me to the following_ https://www.ims.uni-stuttgart.de/forschung/ressourcen/lexika/ghost-nn/
>
> This German compound dataset explicitly focuses on binary compounds with 6 semantic classes. This dataset can not serve as a testbed for our task, as per our previous response. We had done a thorough survey to find potential datasets for the languages exhibiting multi-component compounding phenomenon. However, we could not find such a dataset. We had acknowledged this in our limitation section: “We could not extend our framework to other languages exhibiting multi-component compounding phenomena due to the lack of availability of annotated datasets.”
>
> ---
>
> `Response for questions to authors`
> > _line 251, how do you define "intermediate nodes"? I also do not understand the sentence here ("are stem forms, not categories"). What is a "category"? I don't think this has been mentioned previously._
>
> Due to space constraints, we could not talk about it elaborately. However, at the end of the same section, as per our note (Line 272-274: “We encourage readers to refer to Appendix § A for a more detailed illustration.”), we had given detailed explanations with illustrative examples on it.  Lines 916-923: “Figure 10 illustrates the potential constituency spans for a 3-component compound a−b−c, with ab and bc representing the intermediate compounds. First, the nested structure of compounds does not conform to a syntactic structure; instead, it follows a semantic structure based on the relations between the components rather than their position or relative co-occurrence.” Therefore, it would be helpful if we substitute intermediate nodes ab and bc with their semantic categories. Lines 934-937: “As shown in Figure 11, the intermediate compounds, which lacked additional information, were substituted with their respective semantic compound types in the type identification stage.” We encourage the reviewer to see Figure 10-11 to get a better visualization of it.
>
> > _line 287, would you care to explain what you mean by one-hot span encoding? What are you exactly encoding?_
>
> Thanks for pointing it out. To add more clarity, we will update the final version with the following note: “The span encoding uses two randomly initialized vectors (compound or non-compound) to inform model whether the corresponding instance is compound or non-compound word.” We will rephrase one-hot span encoding as span encoding in our final draft version.
>
> > _line 320-321, please explain the difference between "coarse" and "fine-grained" annotations. Also please provide examples._
>
> We had briefly clarified the key difference in the line 319-321: “We offer two levels of annotations for these datasets: coarse (4 broad types) and fine-grained (86 sub-types).” There are 4 broad semantic types of compounds: Avyayībhava (Indeclinable), Bahuvrīhi (Exocentric), Tatpuruṣa (Endocentric) and Dvandva (Copulative). Again, each broader class is divided into multiple subclasses, leading to 86 fine-grained types. We will update the list of fine-grained labels and the examples in our final draft in the Appendix.
>
> > _line 328, what is "pauranic"? While "ayurveda" can be assumed to be understood by most english speakers, the word "pauranic" has not yet entered common knowledge._
>
> Thanks for pointing it out. We will  update the final draft version with its English translation: Epic. Epic is a genre of ancient Indian literature encompassing historical stories, traditions, and legends.
>
> > _Figure 7: can you add a note to try to explain the unexpected spike for length 10?_
>
> We had mentioned in lines 471-472 :”As the number of components increases, the number of examples in each category decreases.” In lines 488-491 “The NeCTIS dataset exhibits an inherent bias towards left branching, as indicated by the nested tree structure. Consequently, all systems display a bias towards left-branching as well.” Therefore, due to the dominance of left-branching instances and increased variance due to less number of instances, a spike is observed in the results. As per your suggestion, we will add this note in our final version of the draft.
>
> > _Section 7 on "Lexical Semantics": I am not sure much of this is really relevant. Can you motivate it better?_
>
> We had mentioned in line 509-510: ”Lexical semantics is a dedicated field focused on word meaning.” In abstract (line 12-13), we had claimed: “To the best of our knowledge, this is the first attempt in the field of lexical semantics to propose this task.” This motivation aimed to contextualize our work within the larger scope of this field, highlighting our contribution to its advancement.

---

### Meta-Review · Area_Chair_fudf · 2023-09-20

**Recommendation:** 3

**Metareview:**

All reviewers appreciate the work that went into creating the dataset and see the importance of studying complex compounds. However, all reviewers would also welcome clearer explanations, especially of the dataset and annotation procedure, which the authors replied to positively in their rebuttal. All reviewers acknowledged the rebuttals and one reviewer even went into lengthy back-and-forth discussions and increased their scores. Average soundness is at 3.3, and excitement at 3, where the reviewer with the lowest confidence scores providing the lowest scores.
One reviewer mentions that the method could be applied to other languages. It is very nice to see a contribution on a less-studied language such as Sanskrit, but other languages could indeed benefit from such analyses as well. The dataset pointed to does not exactly correspond to the authors needs, but perhaps the authors would like to check the links provided at the bottom that are relevant for bracketing English noun compounds, and the accompanying data (just structure, not labeled relations).

https://aclanthology.org/R15-1094.pdf, https://aclanthology.org/W15-0112/, and David Vadas. 2009. Statistical Parsing of Noun Phrase Structure. Ph.D. thesis.

---

### Decision · Program_Chairs · 2023-10-07

**Decision:**

Accept-Findings

**Comment:**

All reviewers appreciate the work that went into creating the dataset and see the importance of studying complex compounds. However, all reviewers would also welcome clearer explanations, especially of the dataset and annotation procedure, which the authors replied to positively in their rebuttal. All reviewers acknowledged the rebuttals and one reviewer even went into lengthy back-and-forth discussions and increased their scores. Average soundness is at 3.3, and excitement at 3, where the reviewer with the lowest confidence scores providing the lowest scores.
One reviewer mentions that the method could be applied to other languages. It is very nice to see a contribution on a less-studied language such as Sanskrit, but other languages could indeed benefit from such analyses as well. The dataset pointed to does not exactly correspond to the authors needs, but perhaps the authors would like to check the links provided at the bottom that are relevant for bracketing English noun compounds, and the accompanying data (just structure, not labeled relations).

https://aclanthology.org/R15-1094.pdf, https://aclanthology.org/W15-0112/, and David Vadas. 2009. Statistical Parsing of Noun Phrase Structure. Ph.D. thesis.